METHODS AND PROTOCOLS
# An Optimized Transformation Protocol for *Escherichia coli* BW3KD with Supreme DNA Assembly Efficiency

Yuqing Yang,[a,b] Menghui Liu,[a] Tianqi Wang,[a] Qian Wang,[a] Huaiwei Liu,[a] Luying Xun,[a,c] Yongzhen Xia[a]

aState Key Laboratory of Microbial Technology, Shandong University, Qingdao, People's Republic of China
bInstitute of Marine Science and Technology, Shandong University, Qingdao, People's Republic of China
cSchool of Molecular Biosciences, Washington State University, Pullman, Washington, USA

**ABSTRACT** DNA cloning requires two steps: the assembly of recombinant DNA molecules and the transformation of the product into a host organism for replication. High efficiencies in both processes can increase the success rate. Recently, we developed an *Escherichia coli* BW3KD strain with higher transformation efficiency than commonly used cloning strains. Here, we further developed a simple method named TSS-HI (transformation storage solution optimized by Hannahan and Inoue method) for competent cell preparation, which combined the advantages of three common methods for operational simplicity and high transformation efficiency. When competent BW3KD cells were prepared using this developed method, the transformation efficiency reached up to $(7.21 \pm 1.85) \times 10^9$ CFU/$\mu$g DNA, which exceeded the levels of commercial chemically competent cells and homemade electrocompetent cells. BW3KD cells formed colonies within 7 h on lysogeny broth agar plates, quicker than the well-known fast-growing *E. coli* cloning strain Mach1. The competent cells worked effectively for the transformation of assembled DNA of 1 to 7 fragments in one step and promoted efficiencies of transformation or cloning with large plasmids. The cloning efficiency of BW3KD cells prepared by this method increased up to 828-fold over that of *E. coli* XL1-Blue MRF' cells prepared by a common method. Thus, competent cells are suitable for different cloning jobs and should help with the increased demand for DNA assembly in biological studies and biotechnology.

**IMPORTANCE** DNA transformation is commonly used in cloning; however, high transformation efficiency becomes a limiting factor in many applications, such as the construction of CRISPR and DNA libraries, the assembly of multiple fragments, and the transformation of large plasmids. We developed a new competent cell preparation method with unmatched transformation efficiency. When the BW3KD strain, derived from *Escherichia coli* BW25113 cells, was prepared by this method, its transformation efficiency reached up to $(7.21 \pm 1.85) \times 10^9$ CFU/$\mu$g DNA, which broke the record for chemically prepared competent cells. Routine cloning could be completed in 1 day due to the high growth rate of this strain. The competent cells were shown to be highly efficient for transformation or cloning with large plasmids and for the assembly of multiple fragments. The results highlight the effectiveness of the new protocol and the usefulness of the BW3KD strain as the host.

**KEYWORDS** DNA assembly, *Escherichia coli*, chemical competent cells, cloning, transformation

Address correspondence to Yongzhen Xia, xiayongzhen2002@sdu.edu.cn.

The authors declare no conflict of interest.

Restriction enzyme-based cloning is a milestone in molecular biology (1, 2). Recently, restriction enzyme-independent DNA assemblies with short homologous ends have been adapted to reduce experimental difficulties and simplify experimental processes (3, 4). The assemblies usually involve two processes: one is the assembly of recombinant DNA molecules, and the other is the transformation of the product into a host organism. Optimization of either process will facilitate cloning efficiency.

Considerable progress has been made in the optimization of DNA assembly. Numerous DNA assembly methods have been developed (5–8). The Gibson method is a representative method that has contributed to the establishment of synthetic biology (9). Several simplified versions of the Gibson method, including the TEDA (T5 exonuclease-dependent assembly) method, have been developed (5, 10). The goal is to assemble DNA molecules with improved cloning efficiency and simplicity.

*Escherichia coli* remains the most frequently used host for transformation (11). Chemical transformation and electroporation are the two main methods to prepare *E. coli* cells for transformation (12). High transformation efficiency (TE), low cost, and simple preparation procedure are important factors when evaluating competent cell preparation methods. Competent cells with high TE are commonly favored since they increase cloning efficiency, especially for simultaneous cloning of multiple fragments and DNA library constructions (10, 13).

Various methods have been developed to prepare competent cells for transformation to improve TE (Table 1). Although electroporation can introduce up to $10^{10}$ CFU/$\mu$g DNA in the supercoiled plasmid form (14), it is not frequently used with routine cloning methods because the assembled DNA is often purified before electroporation (11). DNA loss during purification and cell damage from electroporation may lead to low cloning efficiency (15). Commercial competent cells normally have a higher TE than homemade competent cells (16). One Shot OmniMAX 2 T1R (Thermo) and XL2-Blue MRF' (Stratagene) are the two commercial competent cell lines with the highest TE ($>5 \times 10^9$ CFU/$\mu$g DNA with supercoiled plasmids), but their expensive prices limit their widespread use, especially by labs with budget constraints. Since their preparation methods are trade secrets, we cannot generate chemically prepared competent cells with a similar TE in the lab. The Hanahan and Inoue methods are two well-known lab-made methods for achieving high TE (17). According to a previous report, competent cells prepared by the Hanahan method may reach $10^9$ CFU/$\mu$g DNA with supercoiled plasmids (16). Although the Inoue method is reported to be better (18), the TEs of both methods are normally maintained at $10^8$ CFU/$\mu$g DNA under standard experimental conditions (19). Competent cells can also be prepared in a transformation storage solution; this method is named the TSS method. The TSS method is a simple one-step procedure for the preparation of competent cells (20). The $1\times$ KCM buffer (0.1 M KCl, 30 mM CaCl$_2$, 50 mM MgCl$_2$) is introduced into the transformation step to enhance the TE in the TSS method (21). Normally, its transformation efficiency is maintained at $\sim10^7$ CFU/$\mu$g DNA. Recently, nanomaterials have been used to increase the TE of *E. coli* (22, 23). However, expensive reagents and time-consuming preparation methods hinder their wide application in most biology labs. Considering the shortcomings of existing methods, the development of an easier and more efficient method to prepare lab-made *E. coli* competent cells will be helpful for cloning jobs.

Our recent work found that the *E. coli* BW25113 strain and its derived strain BW3KD, prepared by using the TSS method, had high TE, reaching $10^9$ CFU/$\mu$g of DNA (24). BW3KD cells contain 3 deleted genes: *endA*, *fhuA*, and *deoR*. *endA* deletion facilitates plasmid isolation, *fhuA* deletion prevents phage infection, and *deoR* deletion facilitates the transformation of large plasmids (17, 25–27). Mutation of these three genes could facilitate either DNA transformation or DNA preparation. The TE has exceeded the levels of most lab-made chemically competent cells. Here, by combining the advantages of the three methods, we further optimized the TSS method with BW3KD cells as the host. The TE of competent cells prepared by this new method was $\sim4.3$- to 19-fold higher than that of the original TSS method for commonly used *E. coli* cloning strains. Importantly, the TE of BW3KD cells prepared by the improved method reached $(7.21 \pm 1.85) \times 10^9$ CFU/$\mu$g DNA. This efficiency exceeded those of commercial products.

## RESULTS

**Improving the TE by optimizing the TSS method with BW3KD cells as the host.** The reagent formulation and method used in TSS were optimized for the BW3KD

**TABLE 1** Main features of competent cell preparation methods

| Prepn method[a] | Feature(s) | E. coli strains used[b] | Typical TE(s) (CFU/$\mu$g)[c] | Reference or source |
|---|---|---|---|---|
| **Chemical methods** | | | | |
| Hanahan's method | (i) Recommended by *Molecular Cloning: a Laboratory Manual*; (ii) purity of reagents and cleanliness of glassware and plasticware affected TE; (iii) few homemade competent cells exceed $10^8$ CFU/$\mu$g of plasmid DNA | DH1, MM294, JM108/9, DH5$\alpha$, DH10B, TOP10, and Mach1 | $10^6$–$10^9$ | 19 |
| Inoue's method | (i) Recommended by *Molecular Cloning: a Laboratory Manual*; (ii) cells needed to be cultured at 18°C | DH5$\alpha$ (typical) and XL1-Blue | $5 \times 10^7$–$3 \times 10^8$ | 36 |
| TSS | (i) Heat shock not necessary for transformation; (ii) PEG 3350 used to improve transformation efficiency | DH5$\alpha$ and DB3.1 | $\sim 5 \times 10^6$ | 21, 33 |
| TSS-HI | (i) Highest TE for chemically competent cells; (ii) other features similar to TSS, except use of heat shock and $MnCl_2$ | BW25113 and its derived strains | $\sim 7 \times 10^9$ | This study |
| Commercial competent cells | (i) Some cells prepared by Hanahan method, but prepn methods not described by most manufacturers | Refer to product catalog | $1 \times 10^9$–$5 \times 10^9$ | Thermo/NEB/Promega/ Sigma |
| **Nanomaterial methods** | | | | |
| Yuan's method | More than 8 steps and more than 9 h needed to prepare nanocatalyst | DH5$\alpha$ | $3.53 \times 10^9$ | 22 |
| Deshmukh's method | (i) More than 15 steps and more than 15 h needed to prepare nanoparticle-DNA complex; (ii) expensive and toxic reagents needed | DH5$\alpha$ | $\sim 10^9$ | 23 |
| **Electroporation** | | | | |
| Classical electroporation method | (i) Recommended by *Molecular Cloning: a Laboratory Manual*; (ii) expensive electroporation equipment needed; (iii) DNA reaction mixtures should be desalted for best TE | DH5$\alpha$, ElectroMAX DH5$\alpha$, Turbo electrocompetent DH10B | Homemade, $>10^9$; commercial, $\sim 10^{10}$ | 44 |
| Commercial competent cells | Detailed prepn methods have not been described | Refer to product catalog | $>1 \times 10^{10}$ | Thermo/NEB/Promega/ Sigma |

[a]TEs higher than $10^9$ CFU/$\mu$g were considered high TEs.
[b]Only strains listed in the reference paper were recorded. Other cloning strains could also be used to prepare competent cells with high TE unless the strain was defined.
[c]Recorded TEs have been mentioned in the associated source or reference.

strain by adjusting 15 factors adopted from the TSS, Hanahan, and Inoue methods (Table 2 and see Fig. S1 in the supplemental material). Cultured cells that were grown to an optical density at 600 nm ($OD_{600}$) of 0.55 and concentrated to 50× had the highest TEs (Fig. S1B and C). The heat shock treatment increased the TE 2-fold (Fig. 1A). Heat shock between 45 and 90 s offered an increased TE (Fig. S1E). Surprisingly, competent cells frozen at −80°C offered improved TE, and quickly freezing cells with liquid nitrogen before storage at −80°C further increased the TE (Fig. 1A).

Metal ions, such as $Ca^{2+}$, $Mn^{2+}$, and $Ru^+$, were added to the TSS buffer to try to improve TE. To prevent the precipitation of $CaSO_4$, $MgSO_4$ in the TSS buffer was changed to $MgCl_2$. The change required a washing step with heat-inactivated TSS (TSS-HI) buffer to maintain TE (Fig. S1H). Different concentrations of $Ca^{2+}$, $Mn^{2+}$, and $Ru^+$ were tested (Fig. S1I to K), and only 140 mM $Mn^{2+}$ increased the TE by $\sim$2-fold (Fig. 1A).

**TABLE 2** Summary of the optimized TSS-HI conditions

| Parameter | Result for: | | Method |
| --- | --- | --- | --- |
| | Tested condition(s) | Optimal condition(s) in TSS-HI | |
| KCM | With or without KCM | With KCM | TSS |
| PEG | PEG 3350 or PEG 8000 | PEG 3350 | TSS |
| Concn of PEG 3350, % | 6, 8, 10, 12, 14 | 10 | TSS |
| Growth stage of cells, $OD_{600}$ | 0.35, 0.55, 0.8, 1.3 | 0.55 | TSS |
| Multiple of cells to concentrate, $\times$ | 1, 10, 50, 100, 200 | 50 | TSS |
| Amt of plasmid used, pg | 10, 50, 200, 1,000 | 200 (pSK-) | TSS |
| Heat shock at 42°C for 90 s | With or without heat shock | With heat shock | Hanahan |
| Heat shock time, s | 30, 45, 90, 120, 180, 240, 300, 360, 720 | 45 or 90 | Hanahan |
| Culture temp, °C | 18, 37 | No difference | Inoue |
| Culture medium | LB or SOC | No difference | Hanahan |
| Recovery conditions | LB, LB+$Mg^{2+}$, or SOC | No difference | Hanahan |
| Freezing conditions | Liquid nitrogen; −80°C; liquid nitrogen with −80°C; no freezing treatment | Liquid nitrogen followed by −80°C storage | TSS |
| Cation(s) | $Mn^{2+}$, $Ca^{2+}$, $Ru^+$, no previous cations | 140 mM $Mn^{2+}$ | Inoue |

The interactions of $Ca^{2+}$, $Mn^{2+}$, and heat shock with TE were analyzed. The presence of either $Ca^{2+}$ or $Mn^{2+}$ promoted TE (Fig. 1B), and heat shock further improved TE (Fig. 1B). Intriguingly, the addition of $Ca^{2+}$ (instead of KCM) in the TSS buffer and the addition of $Mn^{2+}$ in KCM (instead of the TSS buffer) did not increase the TE (Fig. 1B).

**Competent BW3KD cells prepared with TSS-HI have a high TE and a high growth rate.** Several commonly used *E. coli* cloning strains were also prepared by the TSS-HI method and the original TSS method. All displayed increased TEs by the TSS-HI method, but the competent BW3KD cells had the highest TE (Fig. 2A). The TEs by the TSS-HI method were 4.3-, 8-, 11-, 18-, and 19-fold higher than those by the TSS method for strains BW3KD, Stbl3, Mach1, XL1-Blue MRF′, and OmniMAX, respectively (Fig. 2A). These results also indicated that the TEs of competent cells prepared by the TSS-HI method were improved independent of the host strains.

The average TE of competent BW3KD cells prepared with the optimized TSS-HI was $7.21 \times 10^9 \pm 1.85 \times 10^9$ CFU/$\mu$g DNA for 11 batches (Fig. 2B), exceeding the TEs of

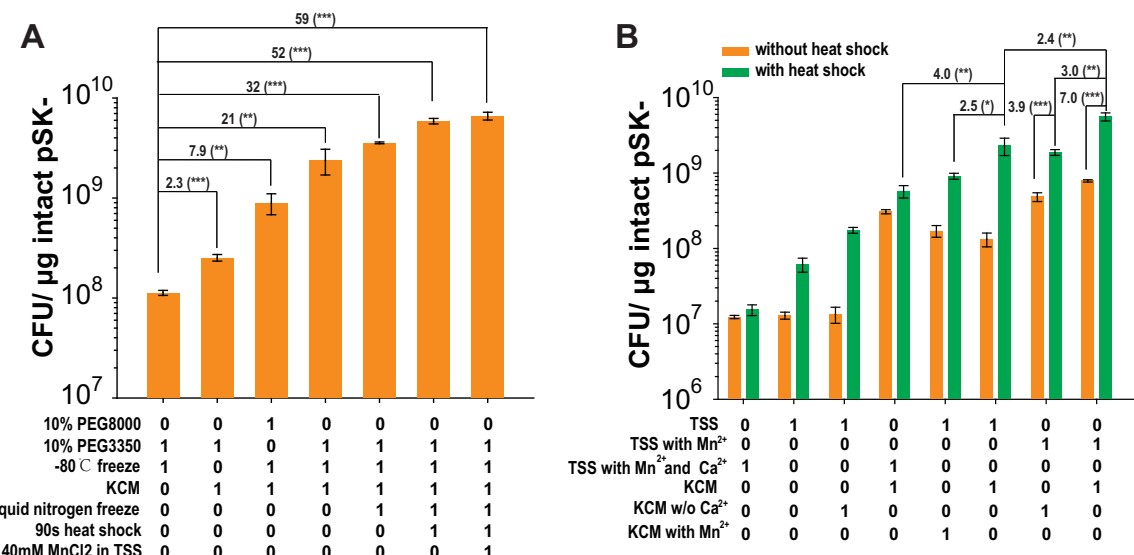

**FIG 1** Optimization of TSS to enhance the TE. (A) The optimization factors to prepare competent BW3KD cells increased the TE. (B) Effects of $Ca^{2+}$, $Mn^{2+}$, KCM (0.5 M KCl, 150 mM $CaCl_2$, 250 mM $MgCl_2$), and heat shock on the TSS-HI method. Data are averages of results from three samples with standard deviations (error bars) ($n$ = 3). One-way analysis of variance (ANOVA) was performed to calculate the $P$ values (*, $P < 0.05$; **, $P < 0.01$; ***, $P < 0.001$). The numbers before the parentheses represent the fold difference. The number 1 shown in the $x$ axis row means the presence of this reagent or the operation, and 0 means the reagent or the operation was not used.

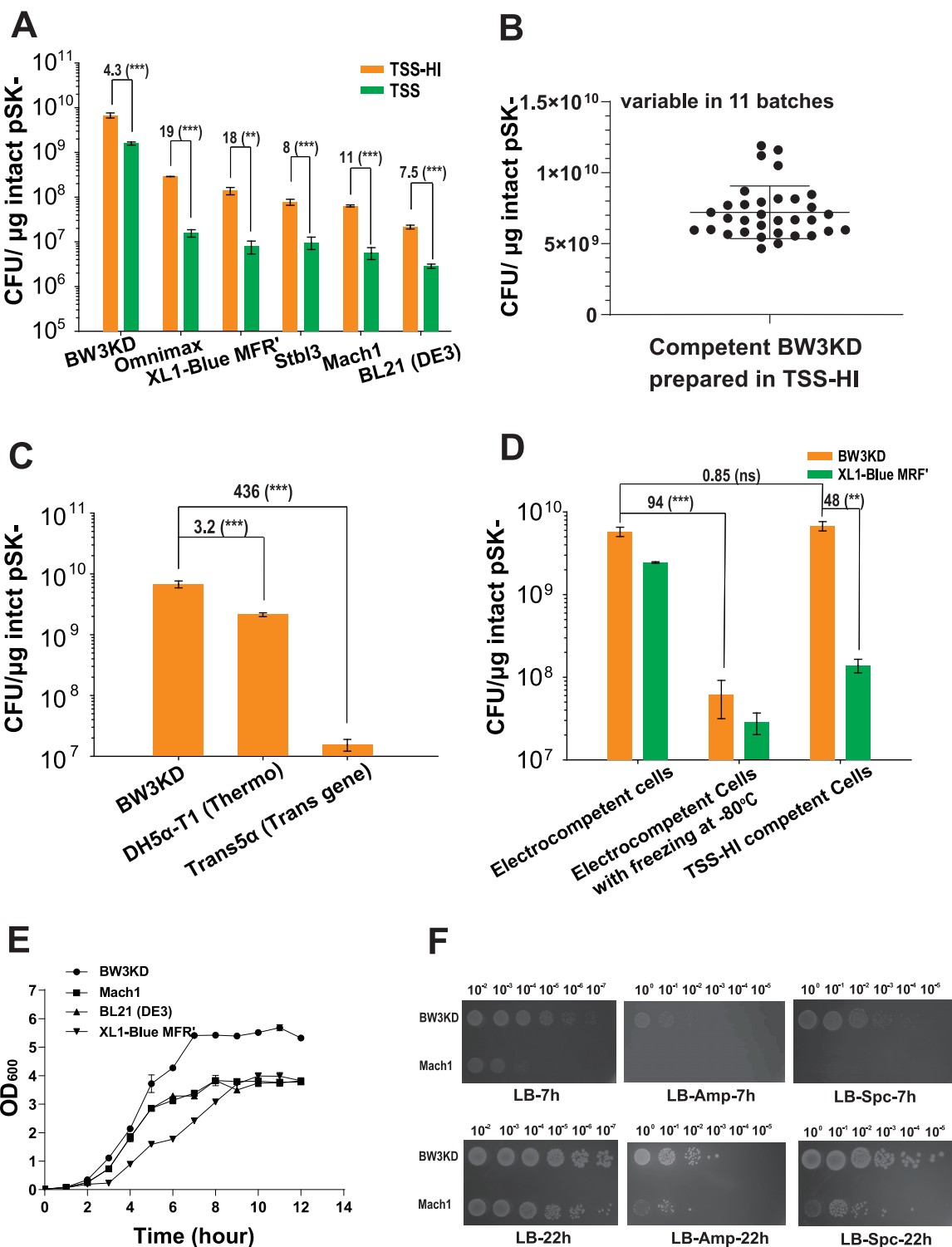

**FIG 2** TEs and growth rates of several *E. coli* strains. (A) TEs of different cloning strains prepared by using TSS-HI and TSS; (B) TEs of the BW3KD competent cells from 11 batches. TSS-HI was used to prepare the competent cells, and the TEs were checked 3 times for each batch. The average of these 33 repeats was calculated to represent the stability of TE. (C) The TE of BW3KD competent cells was compared with the TEs of two commercial strains. (D) Comparison of TEs of BW3KD and XL1-Blue MRF′ cells prepared by TSS-HI and electroporation; (E) growth curves of BW3KD, Mach1, BL21(DE3), and XL1-Blue MRF′ in LB medium at 37°C. (F) The BW3KD and Mach1 strains were made into competent cells by the TSS-HI method and transformed with defined plasmids. After recovery, the cells were diluted and spotted on LB plates with or without defined antibiotics. Photographs were taken at 7 and 22 h after incubation at 37°C. For panels A to E, data are averages from three samples with standard deviations (error bars) (*n* = 3). For panels A and D, unpaired *t* tests were performed (**, *P* < 0.01; ***, *P* < 0.001). For panels C and D, one-way ANOVA was performed to calculate the *P* values. ns, not significant because the *P* value is higher than 0.05. The numbers before the parentheses represent the fold difference.

two commercial competent cells: DH5$\alpha$-T1 (catalog no. 12297016; Thermo Fisher, USA) at 2.14 $\times$ 10$^9$ $\pm$ 1.55 $\times$ 10$^8$ CFU/$\mu$g DNA and Trans5$\alpha$ (catalog no. CD201-01; TransGene Biotech, Beijing, People's Repulic of China) at 1.55 $\times$ 10$^7$ $\pm$ 3.41 $\times$ 10$^6$ CFU/$\mu$g DNA (Fig. 2C). We further compared the TEs of competent BW3KD cells prepared with electroporation and TSS-HI. The TE of BW3KD cells prepared with TSS-HI exceeded that of cells prepared with electroporation (Fig. 2D). Furthermore, the TE of electro-competent cells sharply decreased after the cells were frozen at −80°C (Fig. 2D). In contrast, the competent cells prepared with TSS-HI were more stable after freezing at −80°C and could be stored at −80°C for 3 months without loss of efficiency (Fig. S1O).

We further tested the growth curve of the *E. coli* BW3KD strain with that of the fastest-growing *E. coli* cloning strain, Mach1. In LB medium, BW3KD cells grew faster and accumulated more biomass than Mach1 (Fig. 2E). BW3KD and Mach1 cells were made into competent cells by using the TSS-HI method, and both the recovered transformants and total cells were determined on LB plates with or without appropriate antibiotics. The colonies of BW3KD cells appeared in less than 7 h, and the colonies of Mach1 appeared after 10 h (Fig. 2F). Hence, BW3KD cells could be considered the fastest-growing cloning strain.

*E. coli* BL21(DE3) is commonly used to overexpress cloned genes due to the lack of Lon and OmpT proteases (28). When the competent cells of *E. coli* BL21(DE3) were prepared by the TSS and TSS-HI methods, the TEs were 2.87 $\times$ 10$^6$ and 2.1 $\times$ 10$^7$ CFU/$\mu$g DNA, respectively (Fig. 2A). For cell growth, our results showed that BL21(DE3) grew at a similar rate to Mach1, which was higher than the rate of XL1-Blue MRF′ but lower than that of BW3KD (Fig. 2E). In comparison, the TE of commercial competent BL21 (DE3) cells is normally kept between 1 $\times$ 10$^6$ and ~5 $\times$ 10$^7$ CFU/$\mu$g DNA (catalog no. 200131, Agilent, USA; catalog no. EC0114, Thermo, USA; catalog no. C2527I, NEB, USA). Clearly, the TE of *E. coli* BL21(DE3) prepared in TSS-HI could also match that of the commercial *E. coli* BL21(DE3), which is good for transforming intact plasmids but not for direct cloning. Although the TSS-HI method lifted the TE of *E. coli* BL21(DE3), it is still much lower than those of XL1-Blue MRF' and Mach 1 (Fig. 2A). In summary, BL21(DE3) prepared in TSS-HI is not a good candidate for cloning jobs.

**Competent BW3KD cells prepared with TSS-HI facilitated routine DNA cloning.** Various volumes of the TEDA reaction mixture were transformed into BW3KD cells, and the number of recovered cells was counted (Fig. 3A). One microliter of TEDA reaction mixture was the optimal volume. Furthermore, four different cloning methods were used to clone the 1.2-kb Pkat-enhanced green fluorescent protein (eGFP) fragment into 3-kb SmaI-digested plasmid pSK- to form intact pSK::pKat-eGFP with BW3KD competent cells prepared by using TSS-HI. The competent XL1-Blue MRF′ cells prepared by the TSS method were used as the control, as they have a higher TE than other cloning strains and have been widely used in our previous cloning jobs (10, 29). TEDA had the best cloning efficiency with both strains, and the differences in total transformants between these two strains ranged from 53- to 349-fold, depending on the DNA assembly method (Fig. 3B). Without any treatment, the two DNA fragments had very low cloning efficiency (Fig. 3B). These results indicate that BW3KD competent cells prepared by TSS-HI are ideally used with TEDA for DNA assembly.

**Competent BW3KD cells prepared by TSS-HI facilitated multifragment assembly.** Polyhydroxybutyrate (PHB) is a biodegradable polymer that offers many advantages over traditional petrochemically derived plastics (30, 31). A gene cluster containing five *tac* promoters and three genes for PHB synthesis (10) was cut into several fragments, assembled with linearized vector pSK- by using TEDA, and transformed into the two competent cell lines (Fig. 4A and B). Up to 6 fragments were regularly assembled into pSK- by transforming competent BW3KD cells. In contrast, only up to 5 fragments were assembled into pSK- in the control group, with a 40-fold-lower efficiency than that of BW3KD cells (Fig. 4B).

**Competent BW3KD cells prepared by TSS-HI facilitated the transformation and cloning of large plasmids.** The TEs of some cloning strains sharply decrease when a large plasmid is used (32). Competent BW3KD cells prepared by TSS-HI had increased TE for

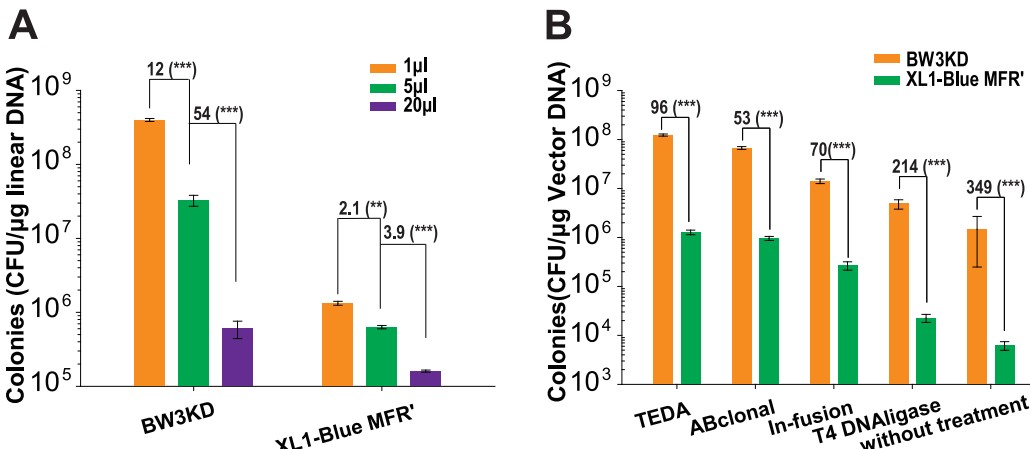

**FIG 3** DNA assembly efficiency after treatment with different cloning methods and transformation into BW3KD and XL1-Blue MRF' cells. (A) The Pkat-eGFP fragment was cloned into pSK- by TEDA, and 1, 5, and 20 $\mu$L of the tested DNA assembly mixtures were transformed into BW3KD or XL1-Blue MRF'. (B) The Pkat-eGFP fragment was cloned into pSK- by using four commonly used cloning methods or without any treatment. These DNA mixtures were transformed into BW3KD or XL1-Blue MRF' cells. The BW3KD competent cells were prepared with TSS-HI, and the XL1-Blue MRF' competent cells were prepared with TSS as the control. Data are averages from three samples with standard deviations (error bars) ($n$ = 3). For panel A, one-way ANOVA was performed to calculate the $P$ values for each data group (*, $P < 0.05$; **, $P < 0.01$; ***, $P < 0.001$). For panel B, unpaired $t$ tests were performed for each pair. The numbers before the parentheses represent the fold difference.

different large plasmids, ranging from 10 kb to 75 kb (Table S1). The results showed that 150-fold to >1,000-fold more transformants were obtained by using BW3KD cells than XL1-Blue MRF' cells (Fig. 5A). Notably, the difference in the number of transformants between the two strains gradually widened as the size of plasmids increased. When 10-, 15-, and 20-kb fragments were assembled with the bacterial artificial chromosome (BAC) plasmid pCC1FOS by using TEDA and transformed into BW3KD, BW2K, and XL1-Blue MRF' cells, respectively, BW3KD cells also showed >100-fold more colonies than the control strain, XL1-Blue MRF' (Fig. 5B). The TE of the BW3KD strain was slightly better than that of

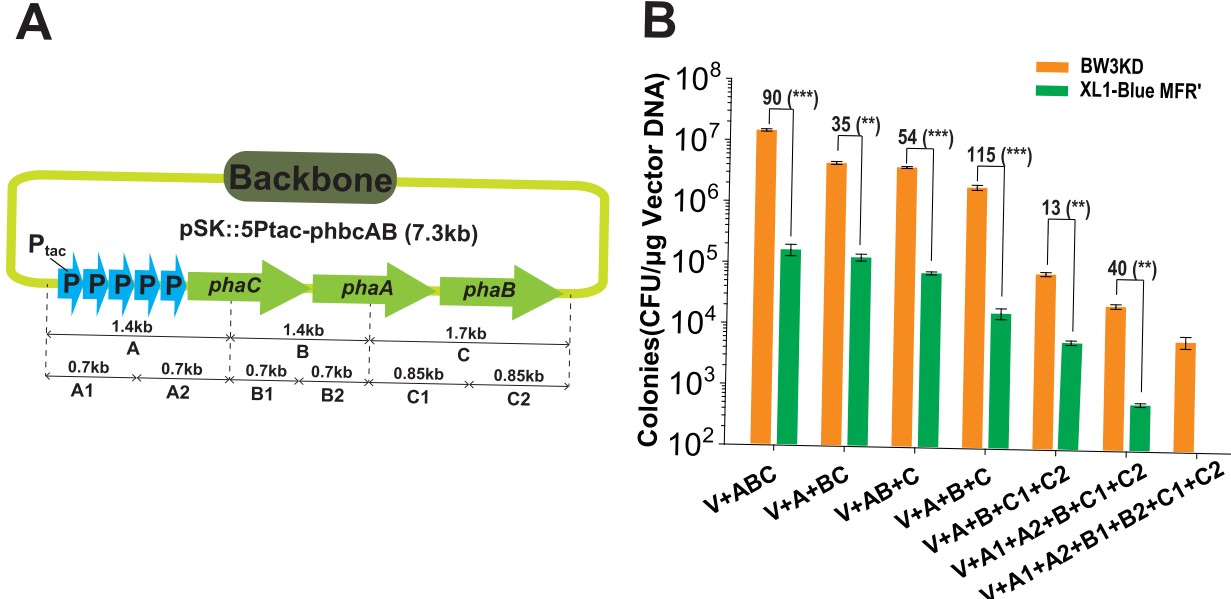

**FIG 4** Multifragment assembly of BW3KD competent cells prepared with TSS-HI. (A) Scheme for assembling multiple fragments using the TEDA method. (B) The total numbers of recovered colonies obtained by assembling multiple fragments with the TEDA method were increased by transforming BW3KD competent cells prepared with TSS-HI. Competent XL1-Blue MRF' cells prepared with TSS were used as the control. Data are averages from three samples with standard deviations (error bars) ($n$ = 3). Unpaired t tests were performed (**, $P < 0.01$; ***, $P < 0.001$). The numbers before the parentheses represent the fold difference.

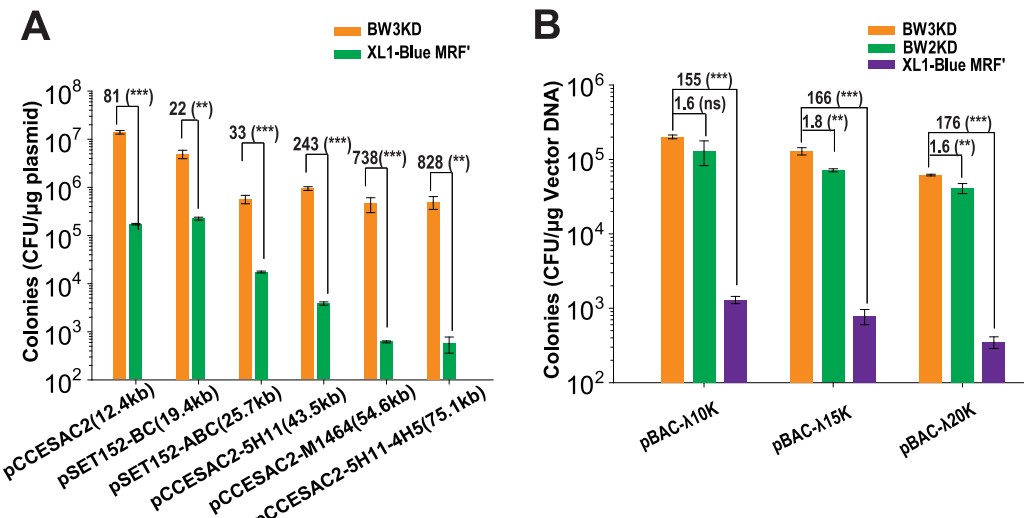

**FIG 5** Large plasmid transformation and cloning with BW3KD competent cells. (A) BW3KD cells prepared with TSS-HI had a high TE when transforming large plasmids (Table S1). Plasmid sizes: pCCESAC2, 12.4 kb; pSET152-BC, 19.4 kb; pSET152-ABC, 25.7 kb; pCCESAC2-5H11, 43.5 kb; pCCESAC2-M1464, 54.6 kb; pCCESAC2-5H11-4H5, 75.1 kb. (B) Large fragments of λ phage DNA were assembled with the BAC plasmid pCC1FOS, and the DNA mixture was transformed into BW3KD and BW2KD competent cells prepared with TSS-HI. XL1-Blue MRF′ prepared with TSS was used as the control for both panels A and B. Data are averages from three samples with standard deviations (error bars) ($n = 3$). Unpaired $t$ tests were performed (**, $P < 0.01$; ***, $P < 0.001$). The numbers before the parentheses represent fold difference.

the BW2K strain with the deletions of *endA* and *fhuA* (Fig. 5B), indicating that the presence of *deoR* may inhibit the transformation of large plasmids as reported (17); however, the inhibition was not profound. Thus, competent BW3KD cells prepared by TSS-HI are especially suitable for transformation and cloning with large plasmids.

## DISCUSSION

A simple chemical method, TSS, was further improved and named TSS-HI to prepare competent cells. The competent *E. coli* BW3KD cells prepared by TSS-HI had the highest TE among chemically prepared competent cells (Fig. 2B and Table 1). The TE of BW3KD cells prepared by using TSS-HI may exceed that of electrocompetent cells (Fig. 2D and Table 1). The TE and cloning efficiencies achieved with BW3KD competent cells prepared by using the TSS-HI method exceeded the levels achieved with XL1-Blue MRF' cells and the competent cells that we previously used by >100-fold and were even higher than those achieved with commercial competent cells (Fig. 2C). BW3KD cells grew as fast as those of the parent strain, BW25113 (24), and they grew even faster than cells of Mach1 (Fig. 2E and F), the fastest-growing *E. coli* strain (19). These results show that the BW3KD strain is a good choice for cloning and that the entire transformation procedure can be performed in 1 day.

The TSS-HI method combines the advantages of the TSS, Hanahan, and Inoue methods. The TSS method is well known for its simplicity (21, 33) and is widely used. The host cells are made competent in one step by incubation with the TSS buffer. Polyethylene glycol (PEG), dimethyl sulfoxide (DMSO), $Mg^{2+}$, and the divalent cations used in TSS buffer are effective in enhancing TE (33). PEG mainly promotes the fusion of DNA and cells (34). For the Hanahan and Inoue methods, the cell competency is partly due to the use of $Ca^{2+}$, $Mn^{2+}$, $Mg^{2+}$, and $K^+$, as well as heat shock treatment. $Ca^{2+}$ and $Mn^{2+}$ reduce the negative charge on the cell surface, promoting the attachment of negatively charged DNA (35). Heat shock may form holes in the cell membrane to facilitate DNA uptake (35). The Hanahan and Inoue methods offer an optimized combination of these factors (19, 36). These factors are also applied in the TSS-HI method. $Ca^{2+}$ and $K^+$ are used in the KCM buffer, $Mn^{2+}$ is included in the TSS buffer (Fig. 1A), and $Mg^{2+}$ is used in both buffers. Addition of $Mn^{2+}$ to KCM did not promote TE

(Fig. 1B). When $Ca^{2+}$ was added to TSS, it did not promote TE (Fig. 1B). Consequently, $Mn^{2+}$ and $Ca^{2+}$ should be included in TSS and KCM, respectively. TSS was previously shown to be the best method to prepare competent BW3KD and BW25113 cells (10). TSS-HI is more effective than TSS (Fig. 2A and 3).

The TSS-HI method is simple, using two buffers and one tube, rivaling many commonly used methods (Table 2). The simplicity of TSS-HI leads to good repeatability (Fig. 2B). In contrast, the TE of lab-made competent cells prepared by the Hanahan and Inoue methods is generally $\sim 10^8$ CFU/$\mu$g DNA (19, 36). Although nanomaterials have been proven to increase the TE to $\sim 10^9$ CFU/$\mu$g DNA (22, 23), the preparation steps are quite time-consuming, and the cost of reagents is high (Table 1). The TE of commercial chemically competent cells is stably maintained between $1 \times 10^9$ and $5 \times 10^9$ CFU/$\mu$g DNA (Table 1), but their prices are too high to be widely used, especially for budget-constrained labs. Here, we disclose a repeatable method that can reach the commercial level of TE by using a simple and economical method to achieve high efficiency.

According to the usage, commercial *E. coli* strains are divided into clonal and protein expression types. Cloning strains generally harbor special characteristics for DNA cloning, including high TE, large plasmid transformation, T1 phage resistance, easy plasmid preparation, and rapid growth (17, 25–27). BW3KD possesses these characteristics for cloning (Fig. 2 and 5).

The commonly used commercial protein expression strain *E. coli* BL21(DE3) has an introduced T7 expression system to achieve protein overexpression (37, 38). BL21 is an *E. coli* B strain that lacks several proteases to prevent the overproduced protein from degradation (28). BW25113 and its mutant BW3KD are of the strain K-12 lineage, and they do not have these characteristics (39). However, BW25113 has been used for heterologous expression in several publications (40, 41). Further efforts are needed to convert BW3KD to a protein expression strain.

In summary, we developed a simple competent cell method, TSS-HI, whose TE had 4.3- to 19-fold increases over the TSS method. Use of the fast-growing strain BW3KD coupled with TSS-HI could result in the highest TE, surpassing the TE of commercial strains. Due to high TE, the competent BW3KD cells prepared with TSS-HI are not only fit for large plasmid transformation and assembly (Fig. 5A and B) but also facilitate gene cloning with either multiple fragments or DNA prepared by different *in vitro* assembly methods (Fig. 4A and B). Because the TE is sufficiently high, direct transformation of untreated DNA fragments can meet the needs for one-fragment cloning (Fig. 3B). Thus, the *E. coli* BW3KD strain prepared with TSS-HI may be applied for diverse DNA cloning jobs.

## MATERIALS AND METHODS

**Strains and plasmids.** The strains used in this study are listed in Table S1 in the supplemental material. All strains were cultured in Luria-Bertani (LB) medium with appropriate antibiotics at 37°C. Ampicillin (Amp), spectinomycin (Spc), Chl, and apramycin (Apr) were used at 100 $\mu$g/mL, 50 $\mu$g/mL, 25 $\mu$g/mL, and 30 $\mu$g/mL, respectively.

**Enzymes and reagents.** Phusion DNA polymerase (Thermo Fisher, USA) was used to amplify DNA fragments. If the size of the DNA fragment was longer than 10 kb, PrimeSTAR GXL DNA polymerase (TaKaRa, Japan) was used. The Trans 2K Plus II DNA marker (TransGen Biotech, Beijing) was used as a DNA ladder to measure the size of DNA fragments by agarose gel electrophoresis. A gel extraction kit, plasmid extraction minikit, and BAC/PAC (P1-derived artificial chromosome) DNA isolation kit (all from Omega, USA) were used to purify DNA. All primers were synthesized by the Beijing Genomics Institute. Magnesium chloride, manganese chloride, rubidium chloride, polyethylene glycol 8000 (PEG 8000), PEG 3350, and DMSO were purchased from Sigma-Aldrich (USA), and the remaining reagents were purchased from Sangon Biotech (Beijing, China).

**Procedures for competent cell preparation and transformation.** To prepare competent cells, a fresh single colony of the defined strain was normally inoculated into 4 mL LB medium and incubated at 37°C overnight. Then, 1% culture was transferred to 50 mL fresh LB medium and cultured at 37°C until an $OD_{600}$ of 0.5 to prepare competent cells for the TSS method and electroporation. The cells were cultured at 20°C to prepare competent cells for the Hanahan and Inoue methods. Cell growth was stopped by incubating the cells on ice for 10 min. Cells were harvested by centrifugation at $4,000 \times g$ and 4°C for 10 min for the preparation of competent cells with different methods.

For the TSS method (21, 33), the harvested cells were resuspended in 1 mL of TSS buffer (LB-HCl [pH 6.1], 10% PEG 3350, 5% DMSO, 10% glycerol, 10 mM $MgSO_4$, 10 mM $MgCl_2$) and chilled on ice for 10 min. This mixture was aliquoted into 30 $\mu$L per tube. Five microliters of 5$\times$ KCM (0.5 M KCl, 150 mM

$CaCl_2$, 250 mM $MgCl_2$) was mixed with DNA and $H_2O$ to a total of 25 $\mu$L of the mixture. The mixture was gently mixed with 25 $\mu$L of competent cells and incubated on ice for 30 min. Then, 250 $\mu$L of fresh LB medium was added for cell recovery at 37°C for 1 h. Recovered cultures were spread onto LB plates with the indicated antibiotics.

For the electroporation method (10), the cells from 50 mL of culture were washed with 20 mL of $H_2O$ twice and 20 mL of 10% glycerol solution once at 4°C, and the washed cells were collected by centrifugation at 4,500 $\times$ g for 10 min. The cells were then resuspended in 600 $\mu$L of 10% glycerol solution and aliquoted into 100 $\mu$L per tube for immediate use. DNA was added and mixed. The mixture was transferred to a sterile 0.2-cm cuvette (Bio-Rad) for electroporation by using a Gene Pulser Xcell electroporation system (Bio-Rad, USA) at 2.5 kV. LB (900 $\mu$L) was added immediately, and the cells were then allowed to recover at 37°C for 1 h. The recovered cultures were spread onto LB plates with the indicated antibiotics.

The TSS-HI method was developed from the TSS method with inputs from the Hanahan and Inoue methods. The detailed procedures are provided in the Materials and Methods section in the supplemental material. The harvested cells were washed once with 1 mL of sterilized TSS-HI buffer (LB-HCl [pH 6.1], 10% PEG 3350, 5% DMSO, 10% glycerol, 20 mM $MgCl_2$, 140 mM $MnCl_2$) and then resuspended in 1 mL of the same buffer as the competent cells, which was aliquoted into 30 $\mu$L per tube for storage or direct use. Five microliters of 5$\times$ KCM (0.5 M KCl, 150 mM $CaCl_2$, 250 mM $MgCl_2$) was mixed with DNA and $H_2O$ to a total of 25 $\mu$L of the mixture. The mixture was gently mixed with 25 $\mu$L of competent cells and incubated on ice for 30 min. The mixture was heat-shocked at 42°C for 90 s and transferred back on ice. Then, 250 $\mu$L of fresh LB medium was added, and the cells were allowed to recover at 37°C for 1 h. The recovered cells were spread onto LB plates with the indicated antibiotics.

All solutions were autoclaved and used under ice-cold conditions. All operations were performed on ice. The cells were gently resuspended in different buffers. After centrifugation, the supernatant was cleaned and quickly pipetted off. The prepared competent cells could be stored at −80°C before use. Competent cells prepared with TSS and TSS-HI should be frozen at −80°C or in liquid nitrogen to promote high TE.

**Preparation of vectors and inserts for DNA assembly.** The plasmids and primers used in this study are listed in Tables S1 and S2, respectively. The TEDA method was used for plasmid construction (10). Briefly, 1 mL of 5$\times$ TEDA solution contained 0.5 M Tris-HCl (pH 7.5), 50 mM $MgCl_2$, 50 mM dithiothreitol, and 0.25 g of PEG 8000. An aliquot of 100 $\mu$L was mixed with 0.1 $\mu$L T5 exonuclease (New England Biolab, Beijing) and diluted to 4/3$\times$ for stock solution. The indicated PCR products and the linearized vector were mixed with 15 $\mu$L of 4/3$\times$ reaction mixture to a final volume of 20 $\mu$L. Between 100 and 200 ng of the linearized vector was used, and the molar ratio of vector to insert was 1:1 to 1:4. The reaction mixture was incubated at 30°C for 40 min before transformation. Competent cells of defined strains prepared by different methods were used to transform either assembled DNA mixtures or intact plasmids as a control.

Four different cloning methods—TEDA (10), Gibson (NEB, US), In-Fusion (TaKaRa, Japan), and restriction ligation—were used to determine if the high TE of the competent cells prepared with TSS-HI could increase the number of total recovered cells on plates. For TEDA, Gibson, and In-Fusion, SmaI-digested pSK- (SmaI-pSK) and the *egfp* gene under the control of the Pkat promoter (Pkat-eGFP) with a 20-bp homology end to SmaI-pSK were used for the DNA assembly assay as previously described (24). For the traditional restriction ligation method, the Pkat-eGFP fragment was amplified to add the SmaI restriction site at both ends. The Pkat-eGFP fragment and pSK- plasmid were restricted with SmaI and ligated with T4 DNA ligase (Thermo, USA). The assembled mixtures were then used for transformation. The Gibson, In-Fusion, and ligation procedures were carried out according to reported methods (42, 43).

TEDA was used for the assembly of multiple DNA fragments. For the assembly of different numbers of DNA fragments, *phbCAB* under the control of five *tac* promoters (5Ptac-phbCAB) from p5TG::phbCAB (24) was amplified via PCR as one fragment (4.3 kb), two fragments (2.8 kb or 1.7 kb and 1.4 kb or 3.1 kb), three fragments (1.4, 1.4, and 1.7 kb), four fragments (1.4, 1.4, 0.85, and 0.85 kb), five fragments (0.7, 0.7, 1.4, 0.85, or 0.85 kb), or six fragments (0.7, 0.7, 0.7, 0.7, 0.85, or 0.85 kb) with 20-bp homologous ends to adjacent fragments or to SmaI-pSK.

The BAC plasmid pCC1FOS was used for the assembly of large fragments. Fragments of 10, 15, and 20 kb were amplified from the $\lambda$ phage DNA to be assembled with the pCC1FOS skeleton by using TEDA. Competent cells prepared by the indicated methods were used for transformation.

**Screening for positive colonies and plasmid stability in cells.** Phenotypes were used for initial screening to determine the positive cloning rates (positive colonies/total colonies). Cells that overexpress *phbCAB* form white colonies on LB plates with 2% glucose due to the accumulation of PHB (14). Colonies producing eGFP were green. Furthermore, 20 colonies were checked by using colony PCR. Ten plasmids from the positive colonies were extracted and checked through agarose gel electrophoresis and DNA sequencing (TsingKe BioTech, People's Republic of China).

**Data availability.** All data are reported in the main text or supplemental material.

## SUPPLEMENTAL MATERIAL

Supplemental material is available online only.

**SUPPLEMENTAL FILE 1**, PDF file, 0.6 MB.

## ACKNOWLEDGMENTS

We thank Haoxin Wang for providing us with the large plasmids for testing, including pCCESAC2, pSET152-BC, pSET152-ABC, pCCESAC2-5H11, pCCESAC2-M1464, and pCCESAC2-5H11-4H5.

We appreciate the support by grants from the National Key Research and Development Program of China (2018YFA0902002), the Natural Science Foundation of China (31961133015 and 31870085), the State Key Laboratory of Microbial Technology, and the Qilu Youth Scholar Startup Funding of SDU (to Y.X.).

We declare no conflict of interest.

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
