## [Reviewer comments · Microbiology Spectrum]

Microbiology Spectrum

An optimized transformation protocol for *Escherichia coli* BW3KD with supreme DNA assembly efficiency

Yuqing Yang, Menghui Liu, Tianqi Wang, Qian Wang, Huaiwei Liu, Luying Xun, and Yongzhen Xia

Corresponding Author(s): Yongzhen Xia, Shandong University

Review Timeline:

Submission Date:	July 3, 2022
Editorial Decision:	August 4, 2022
Revision Received:	September 22, 2022
Editorial Decision:	October 1, 2022
Revision Received:	October 6, 2022
Accepted:	October 14, 2022

Editor: Montarop Yamabhai

Reviewer(s): Disclosure of reviewer identity is with reference to reviewer comments included in decision letter(s). The following individuals involved in review of your submission have agreed to reveal their identity: Friedrich Götz (Reviewer #1)

Transaction Report:

DOI: <https://doi.org/10.1128/spectrum.02497-22>

August 4, 2022

Prof. Yongzhen Xia
Shandong University
State Key Laboratory of Microbial Technology
Qingdao, Shandong 266200
China

Re: Spectrum02497-22 (An optimized transformation protocol for Escherichia coli BW3KD with supreme DNA assembly efficiency)

Dear Prof. Yongzhen Xia:

Thank you for submitting your manuscript to Microbiology Spectrum.

The comments from our two external expert reviewers as you can see below are quite positive. However, there are several important points that should be addressed to improve your manuscript before I can recommend publication in Microbiology Spectrum. So, I would like to ask you to please revise your manuscript accordingly.

While we are willing to consider a revised version of this paper at Spectrum, it would be in your best interest to improve the writing. I recommend that you ask a colleague of yours who is a native English speaker to read and provide you some feedback on the writing. You are also welcome to use one of the services here: <https://journals.asm.org/content/language-editing-services>

Link Not Available

Sincerely,

Montarop Yamabhai

Journals Department
Reviewer comments:

Reviewer #1 (Comments for the Author):

Comment on: An optimized transformation protocol for Escherichia coli BW3KD with supreme DNA assembly efficiency

The authors have developed an E. coli clone, BW3KD, with a higher transformation efficiency than the commonly used cloning strains. Here, they improved the method for competent cell preparation which led to a further increase of transformation efficiency which was also superior to electrocompetent cells. A further advantage of BW3KD was that it formed colonies within 7 hours on lysogeny broth agar plates transformation worked also well with assembled DNA up to 7 fragments and also with large plasmids. The studies with BW3KD were compared with some common E. coli cloning hosts such as Mach1 or E. coli XL1-Blue MRF'.

The development of an E. coli clone with high transformation efficiency and rapid growth is extremely important in molecular biology. However, there are some questions about this paper.

Novelty: In the previous paper BW3KD has been already described and compared with its parent strain BW25113 (Yang et al. 2022, Frontiers in Microbiology). In this paper it was already described that BW3KD had a similar TE to that of BW25113. Here is another comparative description of BW3KD. Nevertheless, the already published work on BW3KD diminishes the novelty of the present work.

Line 61: one that we developed method TEDA, have been developed (5, 10). a) The sentence structure is not correct, b) Although TEDA is described in reference, it should be also briefly described here in M&M.

Fig. 3. Why does increasing DNA content (volume), decrease TE?

Line 168: "colonies of BW3KD could appear in less than 7 hours, but the colonies of Mach1 could not ...". How long does Mach 1 need?

The authors compared BW3KD with E. coli XL1-Blue MRF' and Mach1, but a also very frequent used cloning host, E. coli BL21(DE3), was not compared. Can the authors say something to BL21?

Can the authors say something about the expression of proteins, cytoplasmic and secreted? Many commercial strains are very good in this respect, not only in high TE.

For the readers and users, it would be good if at the end of the manuscript an exact transformation protocol for the competent cell preparation and transformation would be compiled.

BW3KD should be freely available to users at no cost. Also to be able to verify certain properties. This should be guaranteed by authors.

Reviewer #2 (Comments for the Author):

Table 1 under "TSS-HI" "similar to TSS" as a feature is not well explained

If TSS-HI is the name for this new method as in Table 1, then please put this term in the Abstract to facilitate future indexing.

Line 91, the KCM buffer is not explained. What does KCM stand for-it is explained later in methods but would be (briefly) appropriate here as well.

Table 2/Figure S1: Conditions designated as optimal in Table 2 are graphed with error bars in Fig. S1. However, statistical significance of differences between treatments/groups in S1 are not indicated, although visual maxima are easy to spot generally. Some treatment differences with the optimal choices for the TSS-HI procedure indicated in Table 2 are not obviously greatly different in Fig S1. For example, S1, panel E, the chosen conditions from Table 2 show 45 s and 90s being 'equivalent' but the S1 panel E data show 30s as being pretty close to 45s/90s and it is unclear if the results are really significantly lower. Later work in the paper clearly demonstrates that the combined TSS-HI method is highly efficient, so this may not matter in how that total protocol was developed, but if significance were noted, it could help other scientists who come back to modify the TSS-HI protocol later to choose modifications to the protocol without having to see the raw data. Definition of the error bars in the Fig S1 legend would also be needed.

Lines 155-157: It would seem appropriate to me to put suppliers for the commercial competent cells in this discussion?

Line 191: You mention TEDA earlier (line 62) but you did not define it there (other than as a method which is cited), nor here. At line 62 it is probably not necessary to define if it is just a name and the acronym is not important. However, in line 191 those who don't know what the TEDA mixture is may be confused. This should be better explained or spelled out. I think this entire paragraph could deal with some more explanation as the TEDA method is new (2018) and even though it has made a big splash in terms of assembly utility for the cost involved, there are likely many who don't know about it yet. I'd like to see more explanation here.

Line 194: the Pkat-eGFP fragment cloning mentioned should also be better explained-what is the fragment, what is being cloned, etc.

Line 219: I feel like a new section header would be appropriate here to indicate that the work discussed in this paragraph goes beyond the paragraph before.

Line 238: Another section header here is also helpful and appropriate.

Line 314-315: This sentence is a bit unclear and could be explained better.

Line 363-364: Given this is a methods paper, I would love to see a supplemental method or supplemental table that gives an easy to use recipe format for how to make the TSS-HI buffer. While perhaps less traditional in a scientific paper, the ability for scientists to find and use this recipe would be of great interest to the community and be a valuable resource.

Staff Comments:

Preparing Revision Guidelines

Please return the manuscript within 60 days; if you cannot complete the modification within this time period, please contact me. If you do not wish to modify the manuscript and prefer to submit it to another journal, please notify me of your decision immediately so that the manuscript may be formally withdrawn from consideration by Microbiology Spectrum.

Review of “An optimized transformation protocol for Escherichia coli BW3KD with supreme DNA assembly efficiency” by Yang *et al.*

In this paper, the authors report a strain, BW3KD, of *E. coli*, with increased transformation efficiency compared to common cloning strains. High efficiency was maintained with large constructs and with cloning projects involving up to 7 fragments. The protocol to make these competent cells is also reported.

The conclusions of the authors appear to be supported by the data presented. This work is important and could result in benefit to the scientific community by producing higher efficiency competent cells for lower cost, and this will enable transformation of difficult substrates/assemblies as well.

I recommend publication of this work. I do have some specific concerns, noted below.

Table 1 under “TSS-HI” “similar to TSS” as a feature is not well explained

If TSS-HI is the name for this new method as in Table 1, then please put this term in the Abstract to facilitate future indexing.

Line 91, the KCM buffer is not explained. What does KCM stand for—it is explained later in methods but would be (briefly) appropriate here as well.

Table 2/Figure S1: Conditions designated as optimal in Table 2 are graphed with error bars in Fig. S1. However, statistical significance of differences between treatments/groups in S1 are not indicated, although visual maxima are easy to spot generally. Some treatment differences with the optimal choices for the TSS-HI procedure indicated in Table 2 are not obviously greatly different in Fig S1. For example, S1, panel E, the chosen conditions from Table 2 show 45 s and 90s being 'equivalent' but the S1 panel E data show 30s as being pretty close to 45s/90s and it is unclear if the results are really significantly lower. Later work in the paper clearly demonstrates that the combined TSS-HI method is highly efficient, so this may not matter in how that total protocol was developed, but if significance were noted, it could help other scientists who come back to modify the TSS-HI protocol later to choose modifications to the protocol without having to see the raw data. Definition of the error bars in the Fig S1 legend would also be needed.

Lines 155-157: It would seem appropriate to me to put suppliers for the commercial competent cells in this discussion?

Line 191: You mention TEDA earlier (line 62) but you did not define it there (other than as a method which is cited), nor here. At line 62 it is probably not necessary to define if it is just a name and the acronym is not important. However, in line 191 those who don't know what the TEDA mixture is may be confused. This should be better explained or spelled out. I think this entire paragraph could deal with some more explanation as the TEDA method is new (2018)

and even though it has made a big splash in terms of assembly utility for the cost involved, there are likely many who don't know about it yet. I'd like to see more explanation here.

Line 194: the Pkat-eGFP fragment cloning mentioned should also be better explained—what is the fragment, what is being cloned, etc.

Line 219: I feel like a new section header would be appropriate here to indicate that the work discussed in this paragraph goes beyond the paragraph before.

Line 238: Another section header here is also helpful and appropriate.

Line 314-315: This sentence is a bit unclear and could be explained better.

Line 363-364: Given this is a methods paper, I would love to see a supplemental method or supplemental table that gives an easy to use recipe format for how to make the TSS-HI buffer. While perhaps less traditional in a scientific paper, the ability for scientists to find and use this recipe would be of great interest to the community and be a valuable resource.

Point-by-point responses to reviewers

Reviewer #1 (Comments for the Author):

Comment on: An optimized transformation protocol for Escherichia coli BW3KD with supreme DNA assembly efficiency

The authors have developed an E. coli clone, BW3KD, with a higher transformation efficiency than the commonly used cloning strains. Here, they improved the method for competent cell preparation which led to a further increase in transformation efficiency which was also superior to electrocompetent cells. A further advantage of BW3KD was that it formed colonies within 7 hours on lysogeny broth agar plates transformation worked also well with assembled DNA up to 7 fragments and also with large plasmids. The studies with BW3KD were compared with some common E. coli cloning hosts such as Mach1 or E. coli XL1-Blue MRF'.

The development of an E. coli clone with high transformation efficiency and rapid growth is extremely important in molecular biology. However, there are some questions about this paper.

Novelty: In the previous paper BW3KD has been already described and compared with its parent strain BW25113 (Yang et al. 2022, Frontiers in Microbiology). In this paper, it was already described that BW3KD had a similar TE to that of BW25113. Here is another comparative description of BW3KD. Nevertheless, the already published work on BW3KD diminishes the novelty of the present work.

Response: The previous paper focuses on the recognition of BW25113 as an efficient host for DNA assembly. Various mutants were created to understand why it was efficient. BW3KD with the deletion of three genes was not responsible. The key gene was identified as recA, which is normally inactivated in the cloning strains to promote the stability of the cloned genes. This manuscript describes a new method to prepare competent cells with improved TE and showed that BW3KD is suitable for handling large plasmids for efficient DNA assembly. We believe that BW3KD can be a useful host for DNA assembly.

1. Line 61: one that we developed method TEDA, have been developed (5, 10). a) The sentence structure is not correct, b) Although TEDA is described in reference, it should be also briefly described here in M&M.

Response: a) The sentence was changed to "Several simplified Gibson methods, including the TEDA (T5 exonuclease-dependent assembly) method, have been developed" (Line 61-62). b) The brief procedure of TEDA was added in M&M (Line387-396).

2. Fig. 3. Why does increasing DNA content (volume), decrease TE?

Response: When the number of clones was counted by the unit of DNA, it declined (Fig. 3A). We speculated that there are two reasons. First, when a high amount of TEDA-treated DNA was mixed with cells, multiple assembled DNA molecules could enter one cell. Hence, the transformation efficiency per unit of DNA decreased. Second, the

presence of certain ions in the TEDA reaction buffer may reduce the TE of competent cells. The TE of TSS-HI depends on the types of ions and their strengths in the TSS-HI buffer and KCM buffer (Fig. 1A&B). We speculated that excess ions from the TEDA buffer would weaken the TE of competent cells prepared by TSS-HI.

3. Line 168: "colonies of BW3KD could appear in less than 7 hours, but the colonies of Mach1 could not ...". How long does Mach I need?

Response: 10 hours were required for Mach1. That is 2-3 hours more than that required for BW3KD. The information was added in the text (Line 173).

4. The authors compared BW3KD with *E. coli* XL1-Blue MRF' and Mach1, but an also very frequently used cloning host, *E. coli* BL21(DE3), was not compared. Can the authors say something to BL21?

Response: *E. coli* BW25113 belongs to the K-12 lineage, and BL21 is of the strain B lineage. BL21(DE3) is commonly used to overexpress cloned genes due to the lack of Lon and OmpT proteases (Jeong H, et al.; Genome Announc. 2015 Mar 19;3(2):e00134-15.). BL21(DE3) commonly has low transformation efficiency (TE). The TE of commercial competent BL21(DE3) was normally kept between $1 \times 10^6 \sim 5 \times 10^7$ CFU/ μ g DNA (Agilent, Thermo, NEB). When the competent cells of BL21(DE3) were prepared by the TSS-HI method, its TE was 2.1×10^7 CFU/ μ g DNA. It is much lower than XL1-Blue MRF' and Mach 1 (Fig. 2A). For cell growth, our results showed that BL21(DE3) grew faster than XL1-Blue MRF' but similar to Mach1 (Fig. 1 listed in this response letter).

Figure 1. The growth curves of different *E. coli* strains

5. Can the authors say something about the expression of proteins, cytoplasmic and secreted? Many commercial strains are very good in this respect, not only in high TE.

Response: According to the purpose of use, commercial strains are divided into clonal and protein-expression types. Cloning-type strains generally harbor special characteristics for DNA cloning, but rarely have special properties for protein expression (Refer to the links at the end of this paragraph). Apart from high TE, the other main

advantages include easy cloning of large plasmids, T1 phage resistance, easy plasmid preparation, the rapid growth of cells, etc. Our results have proved that BW3KD prepared by TSS-HI not only has high TE but also could clone large plasmids (Fig. 5A&B). In addition, it has the characteristic of rapid growth (Fig. 2E). The knockout of *endA1* has improved the quality of plasmid extraction, and the knockout of the *fluA* gene makes this strain could tolerate T1 phage. Therefore, the BW3KD strain already has comparative advantages in many respects compared with the commercial cloning strains.

Commonly used commercial protein-expression strains generally require the T7 expression system to achieve protein overexpression and purification. Although BW25113 does not yet have this ability, there are well-established methods to engineer *E. coli* strains to acquire a T7 expression system (Tan, et al., ACS Synth. Biol. 2020 Mar 20,9(3):613-622; Hausjell, et al, Microbial Cell Factories, 2018 Oct 30,17(1):169). We have considered upgrading BW3KD to have this ability at a later stage.

BW3KD could overexpress protein with the assistance of native expression systems. BW3KD is derived from BW25113. BW25113 has been used for heterologous expression in many works (Wissner, et al., Journal of Biotechnology, 2021 Jan 10;325:380-388; Xingyan Ma, et al., Front Microbiol., 2020 Jun 3;11:1078). We knocked out *endA1*, *deoR*, and *fluA* of BW25113 to generate BW3KD. The functions of these deleted genes did not affect its heterologous expression ability.

Referred Links for Competent Cell Selection Guide :

1. <https://international.neb.com/tools-and-resources/selection-charts/competent-cell-selection-guide>
2. chrome-extension://efaidnbmnnnibpcajpcglclefindmkaj/https://assets.fishersci.com/TF-S-Assets/LSG/brochures/710_022237_CCGuide_bro.pdf

6. For the readers and users, it would be good if at the end of the manuscript an exact transformation protocol for the competent cell preparation and transformation would be compiled.

Response: A revised detailed protocol was provided in the supplemental section in Line 16-42.

7. BW3KD should be freely available to users at no cost. Also to be able to verify certain properties. This should be guaranteed by the authors.

Response: We will deposit *Escherichia coli* BW3KD to (xx), who will make the strain available to anyone. We make an effort to send the strain to researchers who agree to help us redistribute the strain.

Reviewer #2 (Comments for the Author):

1. Table 1 under "TSS-HI" "similar to TSS" as a feature is not well explained

Response: The sentence was changed to "The other features were similar to TSS except

for the use of heat shock and MnCl_2 " in table 1.

2. If TSS-HI is the name for this new method as in Table 1, then please put this term in the Abstract to facilitate future indexing.

Response: Thanks for your reminder. The TSS-HI was added to the abstract (Line 22).

3. Line 91, the KCM buffer is not explained. What KCM stand for-it is explained later in methods but would be (briefly) appropriate here as well.

Response: It was changed to the sentence "The 1×KCM buffer (0.1 M KCl, 30 mM CaCl_2 , 50 mM MgCl_2) has been introduced into the transformation step to enhance the TE of competent cells prepared with the TSS method" (Line 91-93).

4. Table 2/Figure S1: Conditions designated as optimal in Table 2 are graphed with error bars in Fig. S1. However, the statistical significance of differences between treatments/groups in S1 is not indicated, although visual maxima are easy to spot generally. Some treatment differences with the optimal choices for the TSS-HI procedure indicated in Table 2 are not greatly different in Fig S1. For example, in S1, panel E, the chosen conditions from Table 2 show 45 s and 90s being 'equivalent' but the S1 panel E data show 30s as being pretty close to 45s/90s and it is unclear if the results are significantly lower. Later work in the paper demonstrates that the combined TSS-HI method is highly efficient, so this may not matter in how that total protocol was developed, but if significance were noted, it could help other scientists who come back to modify the TSS-HI protocol later to choose modifications to the protocol without having to see the raw data. A definition of the error bars in the Fig S1 legend would also be needed.

Response: Yes. Statistical significance of differences is important. The statistical significance of differences was added in Fig S1.

5. Lines 155-157: It would seem appropriate to me to put suppliers for the commercial competent cells in this discussion?

Response: The catalog numbers, the brands, and their origins for the commercially competent cells were indicated in the text. (Line 158 – Line 160)

6. Line 191: You mention TEDA earlier (line 62) but you did not define it there (other than as a method that is cited), nor here. In line 62 it is probably not necessary to define if it is just a name and the acronym is not important. However, in line 191 those who don't know what the TEDA mixture is may be confused. This should be better explained or spelled out. I think this entire paragraph could deal with some more explanation as the TEDA method is new (2018) and even though it has made a big splash in terms of assembly utility for the cost involved, there are likely many who don't know about it yet. I'd like to see more explanation here.

Response: Thanks for your suggestion. A detailed explanation for TEDA was added to the text. The full name of TEDA was provided in Line 62. The detailed procedure of TEDA was added in the section on materials and methods (Line 387 – Line 394).

7. Line 194: the Pkat-eGFP fragment cloning mentioned should also be better explained-what is the fragment, what is being cloned, etc.

Response: The explanation was given in Line 195-198. In addition, a more detailed explanation was given in the section on materials and methods (Line 400 – Line 402).

8. Line 219: I feel like a new section header would be appropriate here to indicate that the work discussed in this paragraph goes beyond the paragraph before.

Response: It was added “Competent BW3KD cells prepared with TSS-HI facilitated multi-fragments assembly” in Line 222 as a section header.

Line 238: Another section header here is also helpful and appropriate.

Response: It was added “Competent BW3KD cells prepared with TSS-HI facilitated large DNA transformation and cloning” in Line 243 as a section header.

9. Line 314-315: This sentence is a bit unclear and could be explained better.

Response: The sentence was changed to “Because the TE is high enough, direct transformation of untreated DNA fragments can meet the needs for simple cloning (Fig. 3)” in Line 320-321.

10. Line 363-364: Given this is a methods paper, I would love to see a supplemental method or supplemental table that gives an easy-to-use recipe format for how to make the TSS-HI buffer. While perhaps less traditional in a scientific paper, the ability for scientists to find and use this recipe would be of great interest to the community and be a valuable resource.

Response: The detailed protocol of the TSS-HI method was added in the supplemental section in Line 16-42.

October 1, 2022

Prof. Yongzhen Xia
Shandong University
State Key Laboratory of Microbial Technology
Qingdao, Shandong 266200
China

Re: Spectrum02497-22R1 (An optimized transformation protocol for Escherichia coli BW3KD with supreme DNA assembly efficiency)

Dear Prof. Yongzhen Xia:

It would be nice to put your responses to the reviewer no.1 's comments (no. 4 and 5) to the revised version of the manuscript, as these are the points that should be informed to the readers of Microbiology spectrum as well. As for the response to comment no. 7, please specify what is xx.

Thank you for submitting your manuscript to Microbiology Spectrum. As you will see your paper is very close to acceptance. Please modify the manuscript along the lines I have recommended. As these revisions are quite minor, I expect that you should be able to turn in the revised paper in less than 30 days, if not sooner. If your manuscript was reviewed, you will find the reviewers' comments below.

When submitting the revised version of your paper, please provide (1) point-by-point responses to the issues raised by the reviewers as file type "Response to Reviewers," not in your cover letter, and (2) a PDF file that indicates the changes from the original submission (by highlighting or underlining the changes) as file type "Marked Up Manuscript - For Review Only". Please use this link to submit your revised manuscript. Detailed instructions on submitting your revised paper are below.

Link Not Available

Sincerely,

Montarop Yamabhai

Reviewer comments:

Preparing Revision Guidelines

Please return the manuscript within 60 days; if you cannot complete the modification within this time period, please contact me. If you do not wish to modify the manuscript and prefer to submit it to another journal, please notify me of your decision immediately so that the manuscript may be formally withdrawn from consideration by Microbiology Spectrum.

Point-by-point responses to reviewers

Reviewer #1 (Comments for the Author):

Comment on: An optimized transformation protocol for Escherichia coli BW3KD with supreme DNA assembly efficiency

The authors have developed an E. coli clone, BW3KD, with a higher transformation efficiency than the commonly used cloning strains. Here, they improved the method for competent cell preparation which led to a further increase in transformation efficiency which was also superior to electrocompetent cells. A further advantage of BW3KD was that it formed colonies within 7 hours on lysogeny broth agar plates transformation worked also well with assembled DNA up to 7 fragments and also with large plasmids. The studies with BW3KD were compared with some common E. coli cloning hosts such as Mach1 or E. coli XL1-Blue MRF'.

The development of an E. coli clone with high transformation efficiency and rapid growth is extremely important in molecular biology. However, there are some questions about this paper.

Novelty: In the previous paper BW3KD has been already described and compared with its parent strain BW25113 (Yang et al. 2022, Frontiers in Microbiology). In this paper, it was already described that BW3KD had a similar TE to that of BW25113. Here is another comparative description of BW3KD. Nevertheless, the already published work on BW3KD diminishes the novelty of the present work.

Response: The previous paper focuses on the recognition of BW25113 as an efficient host for DNA assembly. Various mutants were created to understand why it was efficient. BW3KD with the deletion of three genes was not responsible. The key gene was identified as recA, which is normally inactivated in the cloning strains to promote the stability of the cloned genes. This manuscript describes a new method to prepare competent cells with improved TE and showed that BW3KD is suitable for handling large plasmids for efficient DNA assembly. We believe that BW3KD can be a useful host for DNA assembly.

1. Line 61: one that we developed method TEDA, have been developed (5, 10). a) The sentence structure is not correct, b) Although TEDA is described in reference, it should be also briefly described here in M&M.

Response: a) The sentence was changed to "Several simplified versions of the Gibson method, including the TEDA (T5 exonuclease-dependent assembly) method, have been developed" (Line 61-62). b) The brief procedure of TEDA was added in M&M (Line387-396).

2. Fig. 3. Why does increasing DNA content (volume), decrease TE?

Response: TE is the number of colonies produced by transforming 1 ug of DNA (Fig. 3A). More DNA may increase the number of colonies, but not the TE. Further, the presence of certain ions in the TEDA reaction mixture may reduce the TE when increased volume of

the reaction mixture is used.

3. Line 168: "colonies of BW3KD could appear in less than 7 hours, but the colonies of Mach1 could not ...". How long does Mach I need?

Response: 10 hours were required for Mach1. That is 2-3 hours more than that required for BW3KD. The information was added in the text (Line 161).

4. The authors compared BW3KD with E. coli XL1-Blue MRF' and Mach1, but an also very frequently used cloning host, E. coli BL21(DE3), was not compared. Can the authors say something to BL21?

Response: Yes. A paragraph was added (Lines 163-175) and growth curve of BL21(DE3) was added to Fig. 2E.

5. Can the authors say something about the expression of proteins, cytoplasmic and secreted? Many commercial strains are very good in this respect, not only in high TE.

Response: We added one paragraphs on the topics in the discussion (Lines 262-273)

6. For the readers and users, it would be good if at the end of the manuscript an exact transformation protocol for the competent cell preparation and transformation would be compiled.

Response: A revised detailed protocol was provided in the supplemental section in Line 16-42.

7. BW3KD should be freely available to users at no cost. Also to be able to verify certain properties. This should be guaranteed by the authors.

Response: We will make an effort to send the strain to researchers who agree to help us redistribute the strain.

Reviewer #2 (Comments for the Author):

1. Table 1 under "TSS-HI" "similar to TSS" as a feature is not well explained

Response: The sentence was changed to "The other features were similar to TSS except for the use of heat shock and $MnCl_2$ " in table 1.

2. If TSS-HI is the name for this new method as in Table 1, then please put this term in the Abstract to facilitate future indexing.

Response: Thanks for your reminder. The TSS-HI was added to the abstract (Line 21).

3. Line 91, the KCM buffer is not explained. What KCM stand for-it is explained later in methods but would be (briefly) appropriate here as well.

Response: It was changed to the sentence "The 1×KCM buffer (0.1 M KCl, 30 mM $CaCl_2$, 50 mM $MgCl_2$) has been introduced into the transformation step to enhance the TE of competent cells prepared with the TSS method" (Line 90-91).

4. Table 2/Figure S1: Conditions designated as optimal in Table 2 are graphed with error bars in Fig. S1. However, the statistical significance of differences between treatments/groups in S1 is not indicated, although visual maxima are easy to spot generally. Some treatment differences with the optimal choices for the TSS-HI procedure indicated in Table 2 are not greatly different in Fig S1. For example, in S1, panel E, the chosen conditions from Table 2 show 45 s and 90s being 'equivalent' but the S1 panel E data show 30s as being pretty close to 45s/90s and it is unclear if the results are significantly lower. Later work in the paper demonstrates that the combined TSS-HI method is highly efficient, so this may not matter in how that total protocol was developed, but if significance were noted, it could help other scientists who come back to modify the TSS-HI protocol later to choose modifications to the protocol without having to see the raw data. A definition of the error bars in the Fig S1 legend would also be needed.

Response: Yes. Statistical significance of differences is important. The statistical significance of differences was added in Fig S1.

5. Lines 155-157: It would seem appropriate to me to put suppliers for the commercial competent cells in this discussion?

Response: The catalog numbers, the brands, and their origins for the commercially competent cells were indicated in the text. (Line 145 – Line 147)

6. Line 191: You mention TEDA earlier (line 62) but you did not define it there (other than as a method that is cited), nor here. In line 62 it is probably not necessary to define if it is just a name and the acronym is not important. However, in line 191 those who don't know what the TEDA mixture is may be confused. This should be better explained or spelled out. I think this entire paragraph could deal with some more explanation as the TEDA method is new (2018) and even though it has made a big splash in terms of assembly utility for the cost involved, there are likely many who don't know about it yet. I'd like to see more explanation here.

Response: Thanks for your suggestion. A detailed explanation for TEDA was added to the text. The full name of TEDA was provided in Line 62. The detailed procedure of TEDA was added in the section on materials and methods (Line 349 – Line 358).

7. Line 194: the Pkat-eGFP fragment cloning mentioned should also be better explained-what is the fragment, what is being cloned, etc.

Response: The explanation was given in Line 181-184. In addition, a more detailed explanation was given in the section on materials and methods (Line 362 – Line 367).

8. Line 219: I feel like a new section header would be appropriate here to indicate that the work discussed in this paragraph goes beyond the paragraph before.

Response: It was added “Competent BW3KD cells prepared with TSS-HI facilitated multi-fragments assembly” in Line 193 as a section header.

Line 238: Another section header here is also helpful and appropriate.

Response: It was added “Competent BW3KD cells prepared with TSS-HI facilitated the transformation and cloning of large-sized plasmids” in Line 205 as a section header.

9. Line 314-315: This sentence is a bit unclear and could be explained better.

Response: The sentence was changed to “Because the TE is sufficiently high, direct transformation of untreated DNA fragments can meet the needs for one-fragment cloning (Fig. 3B).” in Line 280-282.

10. Line 363-364: Given this is a methods paper, I would love to see a supplemental method or supplemental table that gives an easy-to-use recipe format for how to make the TSS-HI buffer. While perhaps less traditional in a scientific paper, the ability for scientists to find and use this recipe would be of great interest to the community and be a valuable resource.

Response: The detailed protocol of the TSS-HI method was added in the supplemental section in Line 16-42.

October 14, 2022

Prof. Yongzhen Xia
Shandong University
State Key Laboratory of Microbial Technology
Qingdao, Shandong 266200
China

Re: Spectrum02497-22R2 (An optimized transformation protocol for Escherichia coli BW3KD with supreme DNA assembly efficiency)

Dear Prof. Yongzhen Xia:

Congratulations

Your manuscript has been accepted, and I am forwarding it to the ASM Journals Department for publication. You will be notified when your proofs are ready to be viewed.

Sincerely,

Montarop Yamabhai
Editor, Microbiology Spectrum

Journals Department
Supplemental File 1: Accept